# Gamma oscillations in primate primary visual cortex are severely attenuated by small stimulus discontinuities

**Vinay Shirhatti**[1,2¤a], **Poojya Ravishankar**[1¤b], **Supratim Ray**[1,2]*

**1** Centre for Neuroscience, Indian Institute of Science, Bengaluru, India, **2** IISc Mathematics Initiative, Indian Institute of Science, Bengaluru, India

¤a Current address: Department of Neurobiology, University of Chicago, Chicago, Illinois, United States of America
¤b Current address: Committee on Computational Neuroscience, University of Chicago, Chicago, Illinois, United States of America
* sray@iisc.ac.in

**Data Availability Statement:** Data for the figures are located at https://doi.org/10.5281/zenodo.6523772.

**Funding:** This work was supported by Wellcome Trust/DBT India Alliance (IA/S/18/2/504003 –

## Abstract

Gamma oscillations (30 to 80 Hz) have been hypothesized to play an important role in feature binding, based on the observation that continuous long bars induce stronger gamma in the visual cortex than bars with a small gap. Recently, many studies have shown that natural images, which have discontinuities in several low-level features, do not induce strong gamma oscillations, questioning their role in feature binding. However, the effect of different discontinuities on gamma has not been well studied. To address this, we recorded spikes and local field potential from 2 monkeys while they were shown gratings with discontinuities in 4 attributes: space, orientation, phase, or contrast. We found that while these discontinuities only had a modest effect on spiking activity, gamma power drastically reduced in all cases, suggesting that gamma could be a resonant phenomenon. An excitatory–inhibitory population model with stimulus-tuned recurrent inputs showed such resonant properties. Therefore, gamma could be a signature of excitation–inhibition balance, which gets disrupted due to discontinuities.

## Introduction

Gamma oscillations (approximately 30 to 80 Hz) are strongly induced in the primary visual cortex (area V1) by stimuli such as gratings, bars, or colors [1]. One influential hypothesis posits that gamma oscillations play a role in visual perceptual grouping or feature binding, based on the finding that continuous bars induce stronger gamma synchronization between neurons whose receptive fields (RFs) contain parts of the bar as compared to discontinuous bars, even when the discontinuity is outside their RFs [2]. However, in the case of natural images, studies have reported disparate observations regarding the consistency of gamma oscillations [3–5], casting doubts on a causal role for them in feature binding in a natural setting. Because such natural stimuli might occur as discontinuities along many feature dimensions across the RF of neurons, it is important to study how different types of structural irregularities affect firing responses and gamma oscillations. A recent study explored discontinuities in chromatic

Senior Fellowship to S.R and 500145/Z/09/Z – Intermediate fellowship to SR) and DBT-IISc Partnership Programme (to SR). The funders had no role in study design, data collection and analysis, decision to publish, or preparation of the manuscript.

**Competing interests:** The authors have declared that no competing interests exist.

**Abbreviations:** cpd, cycles per degree; ISN, inhibition stabilized network; LFP, local field potential; LR, lateral recurrent; M1, Monkey 1; M2, Monkey 2; PSD, power spectral density; RF, receptive field; TF, time frequency; WSR, Wilcoxon signed rank.

content and showed that stimulus discontinuities can reduce gamma synchronization between responsive neurons [6]. However, the effect of discontinuities along other feature dimensions, such as orientation, phase, and contrast, on gamma oscillations and firing rates remains largely unknown.

In addition, it is unclear how gamma oscillations depend on the size of the discontinuity. Recently, Hermes and colleagues [7] proposed an image-computable model of gamma oscillations, in which the amplitude of gamma depends on the variability across orientation channels. Such a model would also predict a drop in gamma amplitude due to stimulus discontinuities since they can introduce multiple orientations, thereby activating multiple orientation channels and reducing the overall variance across them. Intuitively, reduction in gamma is expected to be graded and proportional to the size of the discontinuity, although since this image-computable model is agnostic to the underlying neuronal network structure and specific network mechanisms, the reduction in gamma with the magnitude of discontinuity could be nonlinear. Other studies have suggested that gamma could be a resonant phenomenon arising due to a tight interplay of excitatory and inhibitory (E-I) signals in a neuronal network [8–12]. Indeed, the V1 RF structure has an excitatory center region flanked by suppressive near-surround and far-surround regions, and involves interactions between feedforward geniculocortical signals, lateral intracortical signals from horizontal connections, and feedback signals from higher areas [13]. Stimulus discontinuities could potentially modulate the interactions between these diverse neuronal subpopulations and alter the levels of E-I in this network, which may result in a drastic reduction in gamma even with a small discontinuity.

To address these questions, we recorded spikes and local field potential (LFP) from area V1 of passively fixating alert monkeys using microelectrode arrays, while they were shown sinusoidal luminance gratings with or without discontinuities that varied along one of 4 dimensions: space, orientation, phase, and contrast. Further, the magnitude of discontinuity for each dimension was parametrically varied. We compared how gamma oscillations and firing rates changed with the magnitude of such discontinuities. Finally, we built an E-I network based on Wilson–Cowan model operating in an inhibition stabilized mode [14–16] and added stimulus-dependent local recurrent inputs to model discontinuities. This simple model could mimic crucial aspect of our observations.

## Results

We implanted a microelectrode array in area V1 of 2 monkeys and estimated the RFs of the recorded sites by flashing small sinusoidal luminance gratings on locations forming a dense rectangular grid on the approximate aggregate RF for all the sites ([17]; see S1 Fig and Materials and methods for details). We have previously shown that large gratings induce 2 distinct gamma oscillations in V1, termed slow (20 to 35 Hz) and fast (35 to 70 Hz) gamma [18]. Here, we presented large static gratings (radii of 9.6˚ and 6.4˚ for the 2 monkeys) at a spatial frequency of 4 cycles per degree (cpd), 100% contrast (except in the contrast discontinuities experiment), at an orientation that induced strong fast gamma oscillations (although for Monkey 2 (M2), these induced moderately strong slow gamma as well), and introduced discontinuities of different types. Therefore, the following results are focused on the fast gamma band, and "gamma" refers to this band. Unless otherwise stated, for the discontinuous gratings, the radius of the inner grating was fixed at 0.3˚ and 0.2˚ for Monkeys 1 and 2, respectively, which was close to the average RF sizes (mean ± SEM for Monkey 1 (M1): 0.28˚ ± 0.009˚, M2: 0.176˚ ± 0.007˚). Thus, the discontinuity across experiments occurred approximately in the visual space corresponding to a transition between the center and surround. In each session, stimuli were centered approximately on the RF center of one of the recorded sites.

## Annular cut discontinuity disrupts gamma

We first tested the effect of a discontinuity in space by introducing an annular cut in the gratings, whose width could take one of the following 5 values: 0˚ (no discontinuity), 0.025˚, 0.05˚, 0.1˚, and 0.2˚ (Fig 1A, topmost row). The trial averaged time frequency (TF) difference spectra for an example site from a session in M1 shows a drastic reduction in gamma power by approximately 58% with an introduction of the smallest tested discontinuity of 0.025˚ (Fig 1A, second row). This effect was qualitatively consistent in the population across multiple sessions in both monkeys (Fig 1A, rows 3 and 4) and was also evident in the average change in power plot (change from baseline period of [–0.5 to 0]s to stimulus period of [0.25 to 0.75]s, where 0 s represents stimulus onset) for different stimuli (Fig 1B). Here, we also observed a second peak at approximately 80 to 100 Hz in M1, which was simply a harmonic of the fast gamma. In comparison, the spiking activity showed only a modest increase across different conditions (Fig 1C) as the surround suppression reduced with increasing annular cut width. We also observed a slight increase in peak frequency with increasing discontinuity (Fig 1B). Such an increase in peak frequency along with a reduction in power has been observed with other stimulus

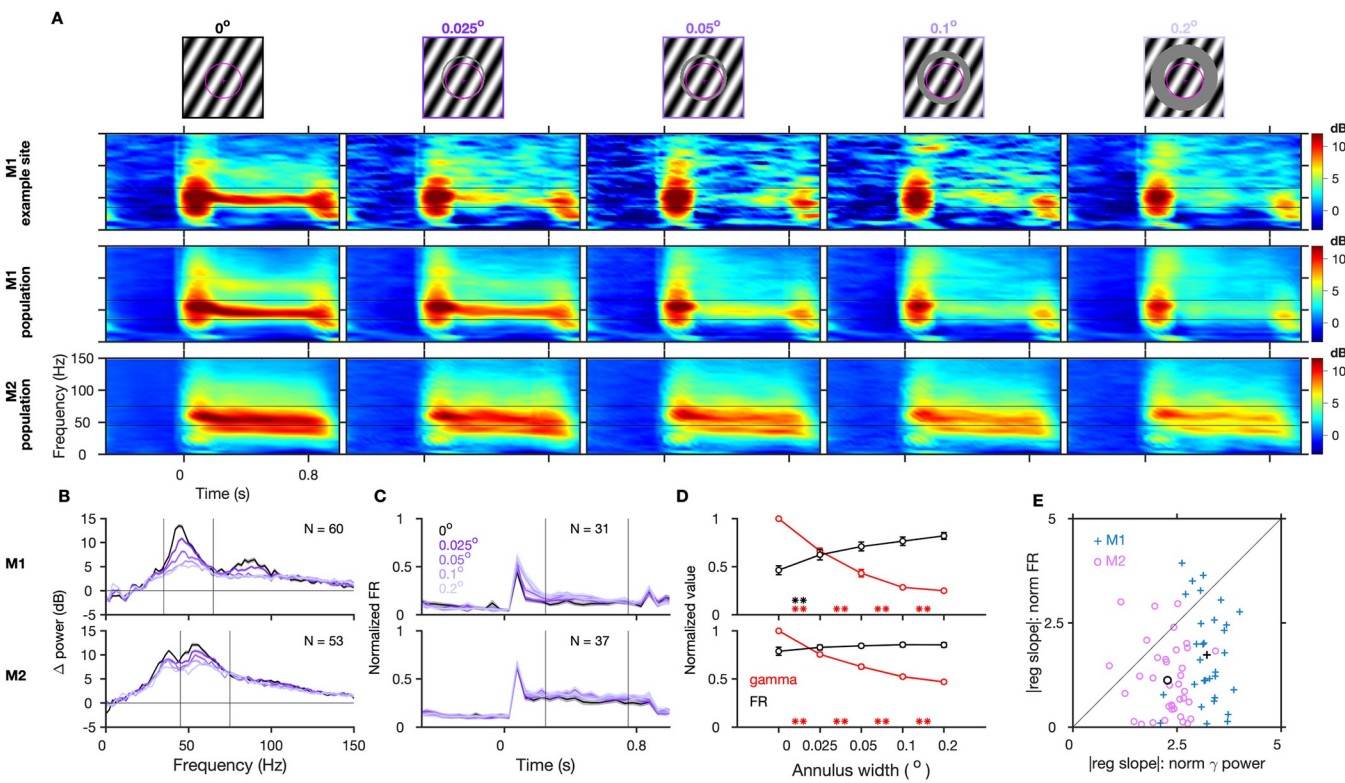

**Fig 1. Gamma oscillations are reduced by annular discontinuity.** (A) Top row: grating stimuli with annular cut discontinuity along with the RF (magenta) of an example center site in M1 from an example session (cut width is mentioned and color coded for other plots). Trial-averaged TF difference spectra for the example center site in M1 (second row), and for the population (averaged across center electrodes and sessions) in M1 and M2 (following rows; number of sites, $N$ = 60 and 53, respectively). Gratings were presented between 0 to 0.8 s. (B) The corresponding mean change in power from baseline ([–0.5 to 0]s) to stimulus period ([0.25 to 0.75]s), averaged across these electrodes and sessions. Black lines in (A) and (B) indicate the gamma ranges (35–65 Hz and 45–75 Hz for the 2 monkeys). (C) Mean normalized firing rate (averaged across selected spiking center electrodes and sessions) for different conditions. (D) Mean normalized gamma power and firing rate during the stimulus period (black lines in (C)) normalized across different annular discontinuity values. The shaded regions and error bars denote the SEM; asterisks in (D) indicate statistical significance (** for $p < 0.01$, * for $p < 0.05$, WSR test) of change in normalized gamma (red) or firing rate (black) between the flanking values of annulus widths. (E) Magnitude of slope of regression with annulus width (Δvalue/degree of visual angle) of normalized gamma power versus normalized firing rate. Each data point represents a selected center electrode; black data points indicate the mean across them. Figure data are located at https://doi.org/10.5281/zenodo.6523772. M1, Monkey 1; M2, Monkey 2; RF, receptive field; SEM, standard error of mean; TF, time frequency; WSR, Wilcoxon signed rank.

manipulations as well, such as with decreasing stimulus size [10,19,20] and superposition of an orthogonal grating to convert a grating to a plaid [21].

To directly compare the relative effects of annular discontinuity on gamma power and spiking activity at the population level, we first selected units that had a mean firing rate of at least 1 spike/s during the stimulus epoch for at least one of the stimulus conditions, normalized both gamma power and firing rates by dividing by the maximum value across the 5 stimulus conditions, and computed the mean across sites (Fig 1D). While the mean normalized gamma power decreased significantly for each discontinuity level in both monkeys, the increase in firing rate was less salient, only reaching significance for the comparison between no discontinuity and the smallest discontinuity in M1. Slopes of regression of change in mean normalized gamma power with increasing annulus width were $-3.22 \pm 0.08$ units/˚ (mean ± SEM units per degree of visual angle) and $-2.27 \pm 0.09$ units/˚ for the 2 monkeys, both significantly negative (Wilcoxon signed rank (WSR) test, M1: z-value = $-4.86$, $p = 0.12 \times 10^{-5}$; M2: z-value = $-5.30$, $p = 0.11 \times 10^{-6}$), whereas similar regression slopes with normalized firing rates were $1.52 \pm 0.25$ units/˚ for M1 (WSR test, z-value = 4.17, $p = 0.3 \times 10^{-4}$) and $0.26 \pm 0.23$ units/˚ for M2 (z-value = 0.82, $p = 0.41$). While gamma power always decreased with the introduction of a discontinuity, the effect on firing rates was more variable. To rule out the possibility that the small overall change in firing rate was not due to larger changes in individual neurons but with opposite signs, we compared the absolute values of the regression slopes for gamma power and firing rates (Fig 1E; the same plot with raw rather than absolute values is shown in S2A Fig). Importantly, the magnitude of slope was significantly greater for gamma power than for firing rates across individual sites (Fig 1E, z-value = 4.37, $p = 0.12 \times 10^{-4}$ and z-value = 4.47, $p = 0.77 \times 10^{-5}$ for M1 and M2, respectively, WSR test), implying that the mean rate of change in normalized gamma with annulus width was significantly greater than that for normalized firing rates. These results indicate that overall gamma was more sensitive to the annular discontinuity than firing rate. We tested whether changes in gamma and firing rates showed any relationship across electrodes, but there was no relationship between the actual regression slopes for gamma and spiking in both the monkeys (see legend of S2 Fig for more details).

To test whether our results depended on the location of the discontinuity, we performed the same experiment with similar discontinuities appearing at one of the following 4 radii from the center of the stimulus: 0.15˚, 0.3˚, 0.6˚, and 1.2˚ in M1 and 0.1˚, 0.2˚, 0.4˚, and 0.8˚ in M2. Gamma was more severely disrupted by a discontinuity that occurred closer to the center (Figs 2A, S3A and S3B). This was reflected in the magnitude of the regression slope between normalized gamma power and annulus width, which decreased for farther cut locations (Fig 2C). On the other hand, increases in firing rate were more modest (Figs 2B and S3C), with regression slopes significantly smaller than the corresponding slopes for gamma for smaller cut radii (Fig 2C).

## Effect of orientation discontinuity

To evaluate the effect of orientation discontinuity, we varied the relative orientation of the inner and outer parts of the static grating stimulus (Fig 3A). Average TF difference spectra across trials of the different stimuli, for an example site from a session in M1, show that gamma oscillations were strongest for matched orientations, and their strength reduced drastically even with the smallest mismatch of 10˚ on both sides (by approximately 72% for a difference of $-10°$ between outer and inner orientation ($(O\text{-}I)°$) and by approximately 69% for $(O\text{-}I)° = 10°$). As this orientation difference systematically increased, there was a drastic reduction in the strength of gamma power across the population in both monkeys as evidenced in the average TF spectra (Fig 3A, rows 3 and 4) and change in power (Fig 3B). As before, a slight

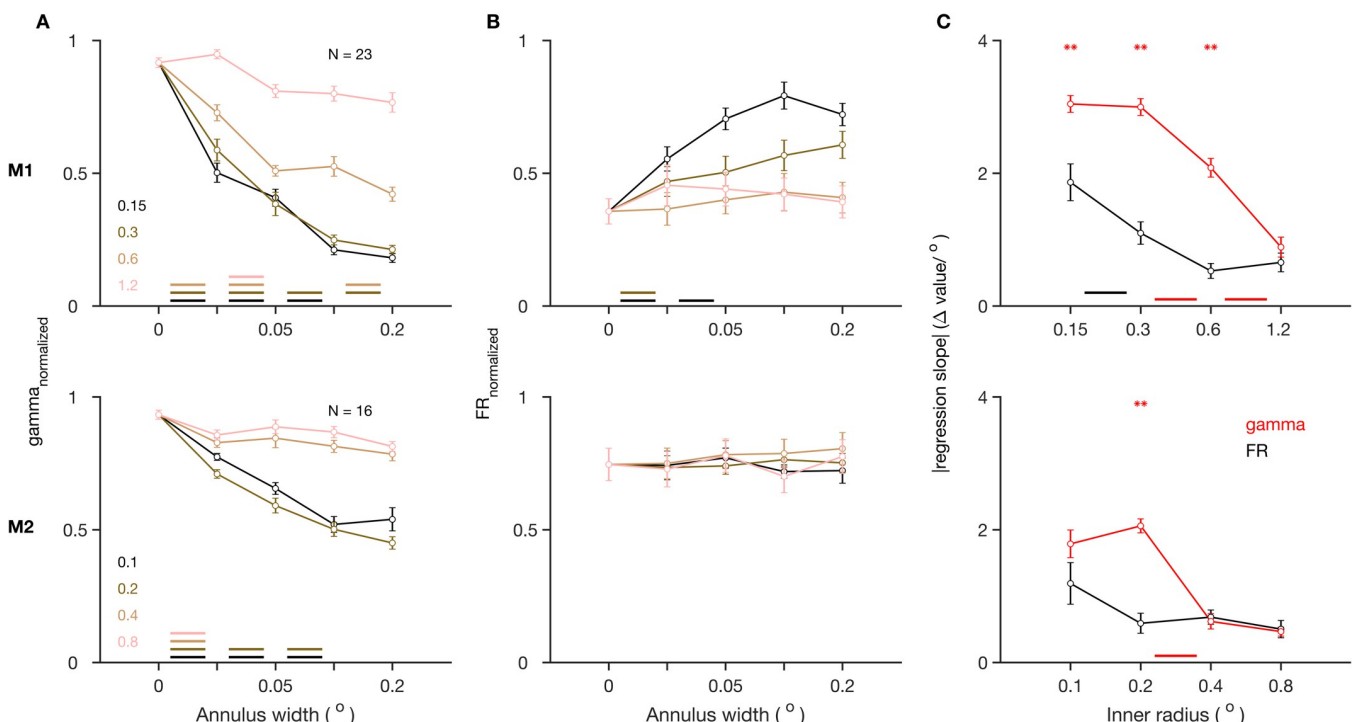

**Fig 2. Effect of annular discontinuity at different locations.** (A) Normalized gamma power and (B) normalized firing rate, for stimuli with annular discontinuity at different inner radii from the center of the stimulus, and as a function of the width of annular discontinuity, averaged across center electrodes, for M1 (top) and M2 (bottom). (C) Magnitude of slope of regression of the values in A and B on annulus width for different inner radii, averaged across the center electrodes. The color-coded lines at the bottom indicate statistical significance ($p < 0.01$, WSR test) of change in corresponding value between the flanking values of annulus widths. The asterisks at the top indicate statistical significance ($p < 0.01$, WSR test) of these values being greater for normalized gamma than for normalized firing rates. Figure data are located at https://doi.org/10.5281/zenodo.6523772. M1, Monkey 1; M2, Monkey 2; WSR, Wilcoxon signed rank.

increase in peak frequency was also observed (Fig 3B). On the other hand, spiking activity increased as the mismatch between inner and outer orientation increased (normalized firing rates, Fig 3C). This is consistent with previous studies of cross orientation suppression where lateral inhibition from the surround has been shown to be the strongest when the center and surround orientation are matched and weakens with increase in orientation differences [22,23].

For a subset of strongly firing units, we compared the dependence on orientation discontinuity of the normalized gamma power to normalized spiking activity (Fig 3D), similar to the analysis in Fig 1D. For the smallest orientation discontinuity that we presented, normalized gamma power reduced significantly for both sides by approximately 48% (for $(O\text{-}I)^\circ = -10^\circ$, z-value = $-4.90$, $p = 0.96 \times 10^{-6}$, WSR test) and approximately 52% (for $(O\text{-}I)^\circ = 10^\circ$, z-value = $-4.94$, $p = 0.80 \times 10^{-6}$) in M1 and approximately 12% (for $(O\text{-}I)^\circ = -10^\circ$, z-value = $-5.40$, $p = 0.67 \times 10^{-7}$) and approximately 15% (for $(O\text{-}I)^\circ = 10^\circ$, z-value = $-6.07$, $p = 0.13 \times 10^{-8}$) in M2, whereas firing rates remained mostly unchanged (Fig 3D, WSR test, z-value = $1.88$, $p = 0.06$ and z-value = $1.06$, $p = 0.29$ for the 2 sides in M1; similarly z-value = $0.47$, $p = 0.64$ and z-value = $-0.78$, $p = 0.43$ in M2). To further compare the rate of change in gamma and firing rates due to orientation discontinuity, we computed the slope of regression of their normalized values with discontinuity on both sides over the range where mean values varied clearly ($(O\text{-}I)^\circ = 0^\circ$ to $\pm20^\circ$ in M1 and $(O\text{-}I)^\circ = 0^\circ$ to $\pm30^\circ$ in M2). Across sites, the slopes were significantly negative for gamma ((z-value = $-4.92$, $p = 0.88 \times 10^{-6}$) and ($-4.94$, $0.80 \times 10^{-6}$) on the 2

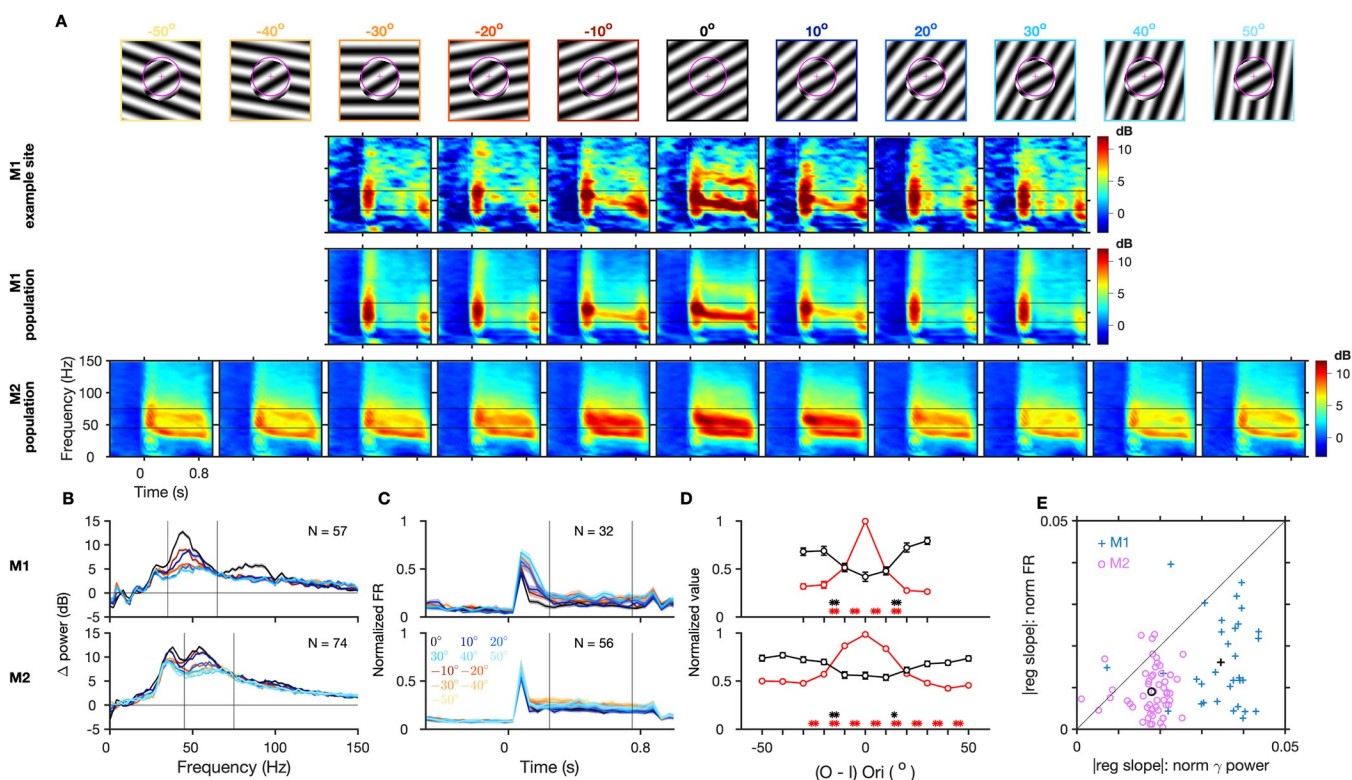

**Fig 3. Gamma oscillations are reduced by orientation discontinuity.** (A–E) Same format as in Fig 1. For M2, there are 2 additional steps of discontinuity on either side (−50˚ to 50˚ in steps of 10˚). Figure data are located at https://doi.org/10.5281/zenodo.6523772. M1, Monkey 1; M2, Monkey 2.

sides in M1 and (−6.51, 0.75 × 10⁻¹⁰) and (−6.48, 0.89 × 10⁻¹⁰) in M2, WSR test) and significantly positive for firing rates ((z-value = 3.78, $p = 0.16 \times 10^{-4}$) and (4.04, 0.54 × 10⁻⁴) in M1 (4, $p = 0.64 \times 10^{-4}$) and (2.82, 0.48 × 10⁻²) in M2, WSR test). However, the magnitude of the mean slope (averaged across both sides) was larger for gamma than for firing rates consistently across individual sites in both the monkeys (Fig 3E, (z-value = 4.60, $p = 0.42 \times 10^{-5}$) in M1 and (5.80, 0.66 × 10⁻⁸) in M2, WSR test). As in the previous case, the mean slopes were consistently negative for gamma across all sites for both monkeys, whereas there was more heterogeneity of effects for spiking responses (S2B Fig). Interestingly, the mean firing responses changed significantly between conditions of 10˚ to 20˚ orientation differences, after which they again remained comparable for increasing orientation differences in both the monkeys (Fig 3D). Gamma, on the other hand, was sensitive to orientation differences across a larger range, and this effect was more consistent across the recorded sites.

## Effect of phase discontinuity

Next, we introduced increasing levels of spatial phase difference between the inner and outer gratings. Gamma dropped drastically with the smallest phase difference of 60˚ on both sides (both $(O-I)\phi˚ = 60˚$ and 300˚ imply a phase difference of 60˚ from 2 sides of the spatial cycle) as seen in the average TF difference spectra for an example site in M1 (Fig 4A, row 2, gamma reduced by approximately 57% for $(O-I)\phi˚ = 60˚$ and approximately 52% for $(O-I)\phi˚ = 300˚$) and for the population in both the monkeys (Fig 4A, third and fourth rows). Across the population, the matched condition showed the strongest gamma oscillations and the strength of gamma reduced and peak frequency increased progressively as the phase disparity increased

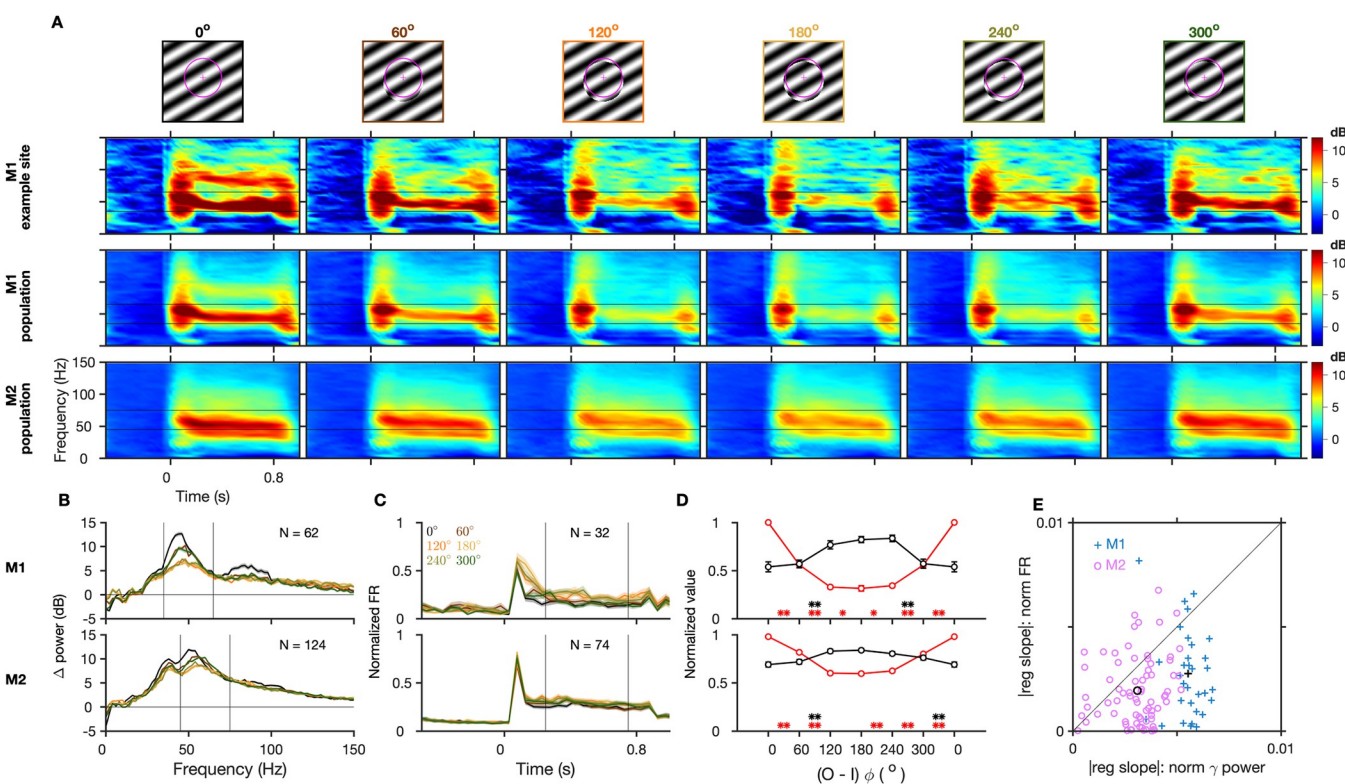

**Fig 4. Gamma oscillations are reduced by phase discontinuity.** (A–E) Same format as in Figs 1 and 3. In (D), the values for 0˚ are repeated on the right side due to the circular scale of phases and for easy comparison with values for 300˚, which is one of the 2 smallest discontinuities in this stimulus set. Figure data are located at https://doi.org/10.5281/zenodo.6523772. M1, Monkey 1; M2, Monkey 2.

from both sides of the spatial cycle until 180˚, at which the inner and outer gratings were in perfect anti-phase, showing that gamma changed in accordance with the magnitude of discontinuity (Fig 4A and 4B). Firing rates remained mostly unchanged with the smallest discontinuities and showed a small increase with bigger discontinuities (Fig 4C and 4D).

The stimulus period mean normalized gamma power (Fig 4D, similar to Figs 1D and 3D) reduced significantly by approximately 44% with the smallest phase differences on both sides in M1 (2-sided $t$ test, $t_{(31)} = -18.23$, $p = 0.40 \times 10^{-17}$ for $(O-I)\phi^\circ = 60^\circ$ and $t_{(31)} = -14.65$, $0.18 \times 10^{-14}$ for $(O-I)\phi^\circ = 300^\circ$) and by approximately 17% $((O-I)\phi^\circ = 60^\circ$, $t_{(73)} = -9.49$, $p = 0.23 \times 10^{-13})$ and approximately 18% $((O-I)\phi^\circ = 300^\circ$, $t_{(73)} = -11.43$, $p = 0.62 \times 10^{-17})$ in M2. The corresponding change in normalized firing rates was not significant in M1 ($t_{(31)} = 0.93$, $p = 0.36$ and $t_{(31)} = 0.71$, $p = 0.49$, 2-sided $t$ test) and increased significantly only for $(O-I)$ $\phi^\circ = 300^\circ$ in M2 ($t_{(73)} = 3.61$, $p = 0.57 \times 10^{-3}$; $t_{(73)} = 1.13$, $p = 0.26$ for $(O-I)\phi^\circ = 60^\circ$). To compare the dependence of gamma and firing rates on phase discontinuity at each site, we computed the slope of regression of their normalized values with phase discontinuities from 0˚ to 180˚ from both sides of the spatial cycle. Slopes were significantly negative for normalized gamma ((z-value = $-4.94$, $p = 0.8 \times 10^{-6}$) for both sides in M1 ($-7.46$, $0.87 \times 10^{-13}$) and ($-7.35$, $0.20 \times 10^{-12}$) for M2, WSR test) and significantly positive for firing rates ((z-value = $4.02$, $p = 0.58 \times 10^{-4}$) and ($3.29$, $0.10 \times 10^{-2}$) in M1 ($3.42$, $0.62 \times 10^{-3}$) and ($3.84$, $0.12 \times 10^{-3}$) in M2) on both sides in both monkeys. The magnitude of slope (averaged across both sides) was significantly larger for gamma than for firing rates across individual sites (Fig 4E, z-value = $4.26$, $p = 0.20 \times 10^{-4}$ in M1; z-value = $4.65$, $p = 0.33 \times 10^{-5}$ in M2; see S2C Fig for raw slope values, which were consistently negative for gamma across all sites in both monkeys, and more

variable for firing rates). Thus, change in gamma power was more sensitive to phase discontinuities for a much broader range compared to firing rate and showed a higher sensitivity consistently across sites. The most significant changes in firing responses were observed for bigger mismatches in phases of 120˚ or 240˚ between the inner and outer region, not increasing further for larger mismatches in both monkeys (Fig 4D, black line). Thus, there is a critical subrange of maximum sensitivity of firing rates to phase differences, similar to the orientation discontinuity effects.

## Effect of contrast discontinuity

Increasing the contrast of a visual stimulus has been shown to increase the spiking responses as well as the strength [24] and frequency of gamma oscillations in V1 [10,25]. Likewise, increasing the stimulus size, which is expected to increase surround suppression, also increases gamma power but reduces the peak frequency [10,19,20]. We studied how a discontinuity in the contrast profile across the inner and outer regions of the grating affects gamma oscillations, by varying their contrast independently over 5 different values (25 combinations). In Fig 5A, the inner and outer contrasts were matched along the diagonal, and contrast discontinuity progressively increased away from this diagonal. If gamma depended mainly on the overall input strength, it should increase from the left to the right column (Fig 5A, as outer contrast increases) and from the top to the bottom row (as inner contrast increases). Instead, gamma was strongest for the contrast matched stimuli along the diagonal compared to stimuli away

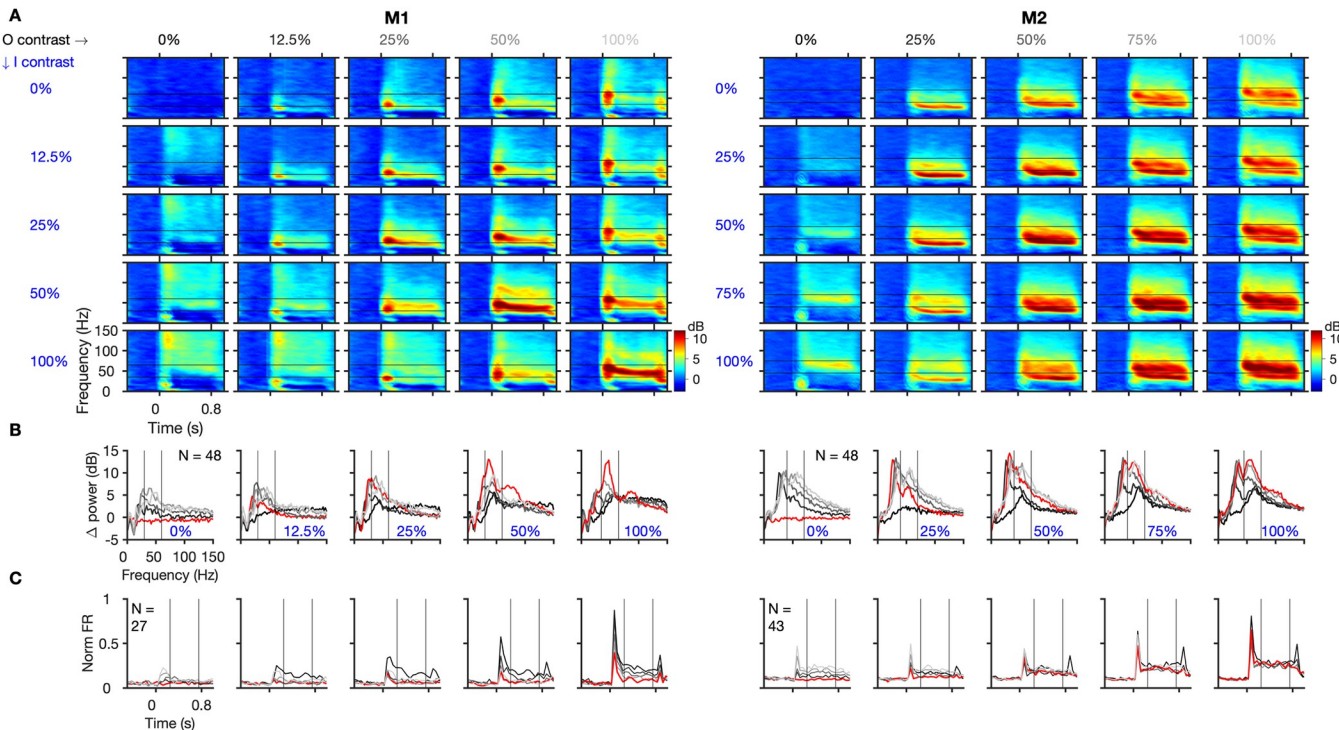

**Fig 5. Gamma oscillations are reduced by contrast discontinuity.** (A) Trial-averaged TF difference spectra for the population, induced by stimuli with contrast discontinuity (inner and outer contrasts are indicated on the left and top, respectively) and (B) the corresponding mean change in power from baseline to stimulus period. (C) Mean normalized firing rate averaged across selected spiking center electrodes and sessions. Same format as Fig 3A–3C. In B–C, every column shows graphs for varying outer contrast (increasing with brightness of gray), at a fixed inner contrast (mentioned in (B)), and the colored red line represents the stimulus with the corresponding matched inner-outer contrast. Figure data are located at https://doi.org/10.5281/zenodo.6523772. M1, Monkey 1; M2, Monkey 2; TF, time frequency.

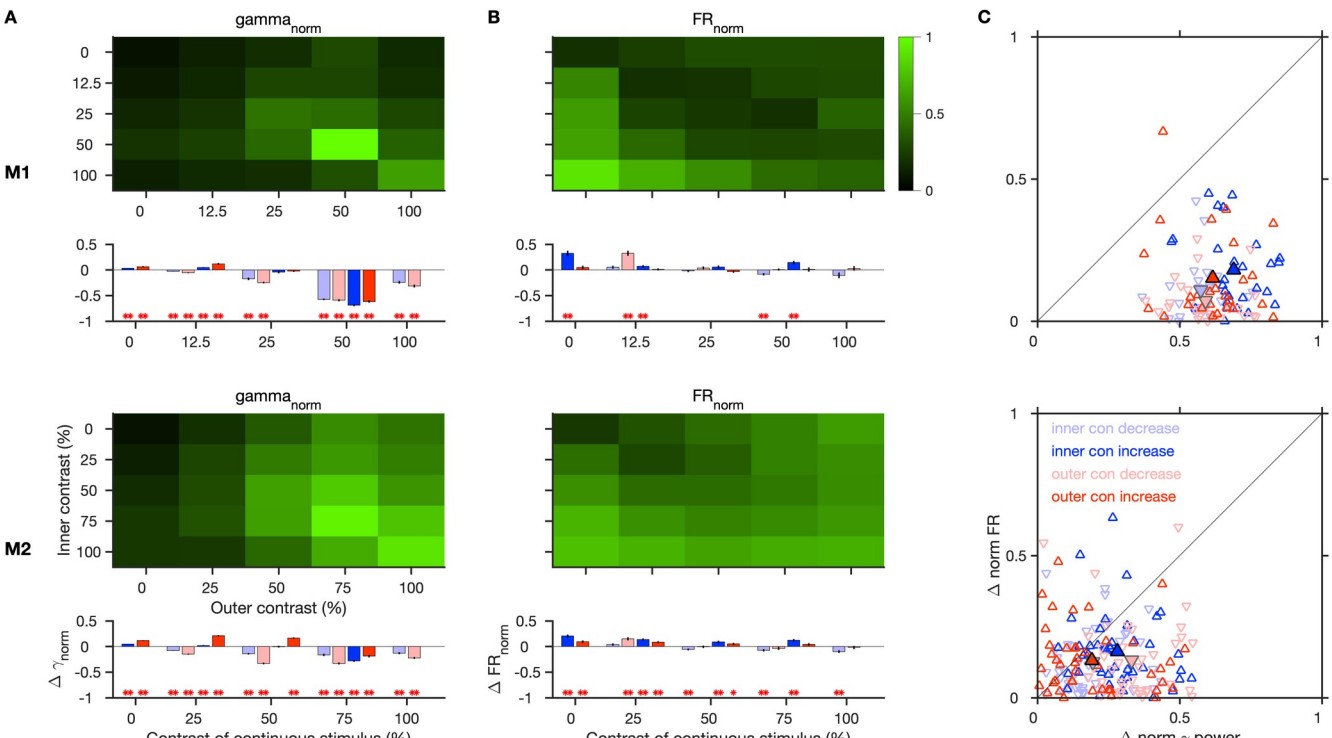

**Fig 6. Effect of contrast discontinuity on gamma and firing responses.** (A) Normalized gamma power and (B) normalized firing rate during the stimulus period for different outer and inner contrast conditions, for the same electrodes as in Fig 5C, for M1 (top) and M2 (bottom). The corresponding effect of change in contrast from a continuous stimulus is shown below each. The bar graphs denote the change for a step change (decrease or increase) in contrast occurring in either the inner or the outer grating, from a continuous stimulus whose contrast is indicated on the x-axis. Asterisks indicate its statistical significance (** for $p < 0.01$, * for $p < 0.05$, WSR test). (C) Magnitude of change in normalized gamma power versus normalized firing rates for the 4 smallest discontinuities from 50% contrast and 75% contrast continuous grating in M1 and M2, respectively. Larger and bordered data points show the mean across sites. Figure data are located at https://doi.org/10.5281/zenodo.6523772. M1, Monkey 1; M2, Monkey 2; WSR, Wilcoxon signed rank.

from the diagonal. This is also clear in Fig 5B, where each plot represents the change in power for a fixed inner contrast and varying levels of outer contrast, and in the comparison of the mean level of normalized gamma for all stimulus conditions, averaged across all center sites across sessions (Fig 6A). To quantify this effect, we compared the gamma power for each contrast matched stimulus and its neighboring mismatched stimuli, obtained by either decreasing or increasing the contrast of either the inner or outer grating only (Fig 6A, bar plots). Gamma rhythm was negligible for contrast matched stimuli at low contrasts (below 25% in our stimulus set) and therefore power in this band simply followed the change in overall stimulus drive due to discontinuities at low contrasts. However, at higher contrasts (50% and above), any change in contrast of either the center or surround (decrease or increase) from the matched case caused a significant reduction in gamma power. This effect was especially stark when the contrast matched stimulus at 50% or 75% (in M2) was changed.

The spiking activity showed a more direct dependence on the contrast, generally increasing with an increase in inner contrast or reduction in outer contrast (Figs 5C and 6B), and decreasing with a decrease in inner contrast. With an increase in outer contrast, the firing rates occasionally increased, especially at low-level center contrasts (0%, 25%) in M2, although this increase was smaller than the increase due to center contrast. Such an effect of surround contrast is not unexpected, since at low contrasts surround has been shown to be facilitatory [26]. At mid and high contrast discontinuities, firing rates mostly remained unchanged.

We compared the effect of contrast discontinuity on gamma and firing rates at each site by plotting the magnitude of change in their normalized values from the continuous case with strongest gamma rhythm (50% in M1 and 75% in M2) to the 4 immediate discontinuity cases, i.e., when either the inner or outer contrast changed by 1 step (Fig 6C). In both monkeys, the change in gamma was significantly greater than for firing rates in almost all cases ((z-value = 4.54, $p = 0.56 \times 10^{-5}$), (4.54, $0.56 \times 10^{-5}$), (4.54, $0.56 \times 10^{-5}$), and (4.47, $0.79 \times 10^{-5}$) in M1 for inner contrast decrease and increase, and outer contrast decrease and increase, respectively; (z-value = 3.13, $p = 0.18 \times 10^{-2}$), (3.86, $0.11 \times 10^{-3}$), (4.83, $0.14 \times 10^{-5}$), and (1.76, $0.78 \times 10^{-1}$) in M2, WSR test). Thus, gamma was more sensitive to contrast discontinuities than firing rates.

## A resonant model of gamma oscillations

To understand the resonant properties of gamma oscillations, we used a previously well-studied lumped neuronal model, consisting of an excitatory (E) and an inhibitory (I) neuronal population that can generate oscillations ([14], Eq 1, refer "Model details" in Materials and methods). To model discontinuities in this framework, we included additional lateral recurrent (LR) inputs (Eqs 2 and 3), which were further dependent on the overall excitatory and inhibitory activity of the population (Fig 7). This effectively changed the weights of the network model (see Materials and methods for details). In this framework, loss of recurrent inputs due to a discontinuity could be modeled as a reduction in some of the weights.

Fig 8A shows the mean firing rates of the E population, I population, peak gamma power, and its frequency, as a function of the external inputs given to the E and I populations (*iE* and

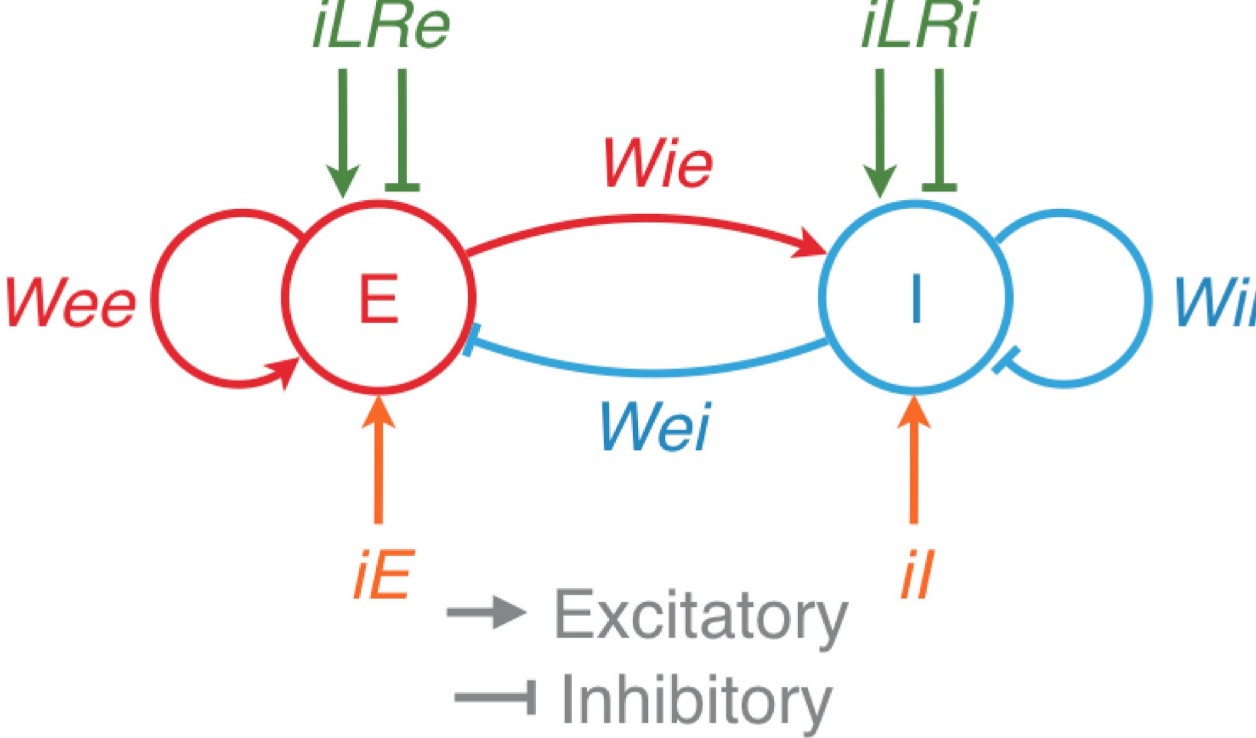

**Fig 7. A resonant model of gamma oscillations.** A schematic of the model with an excitatory population (E) and an inhibitory population (I). *iE* and *iI* are the external inputs, and *iLRe* and *iLRi* are inputs from other nearby local recurrent networks. *Wee*, *Wei*, *Wie*, and *Wii* are the gains of the corresponding connections between the populations.

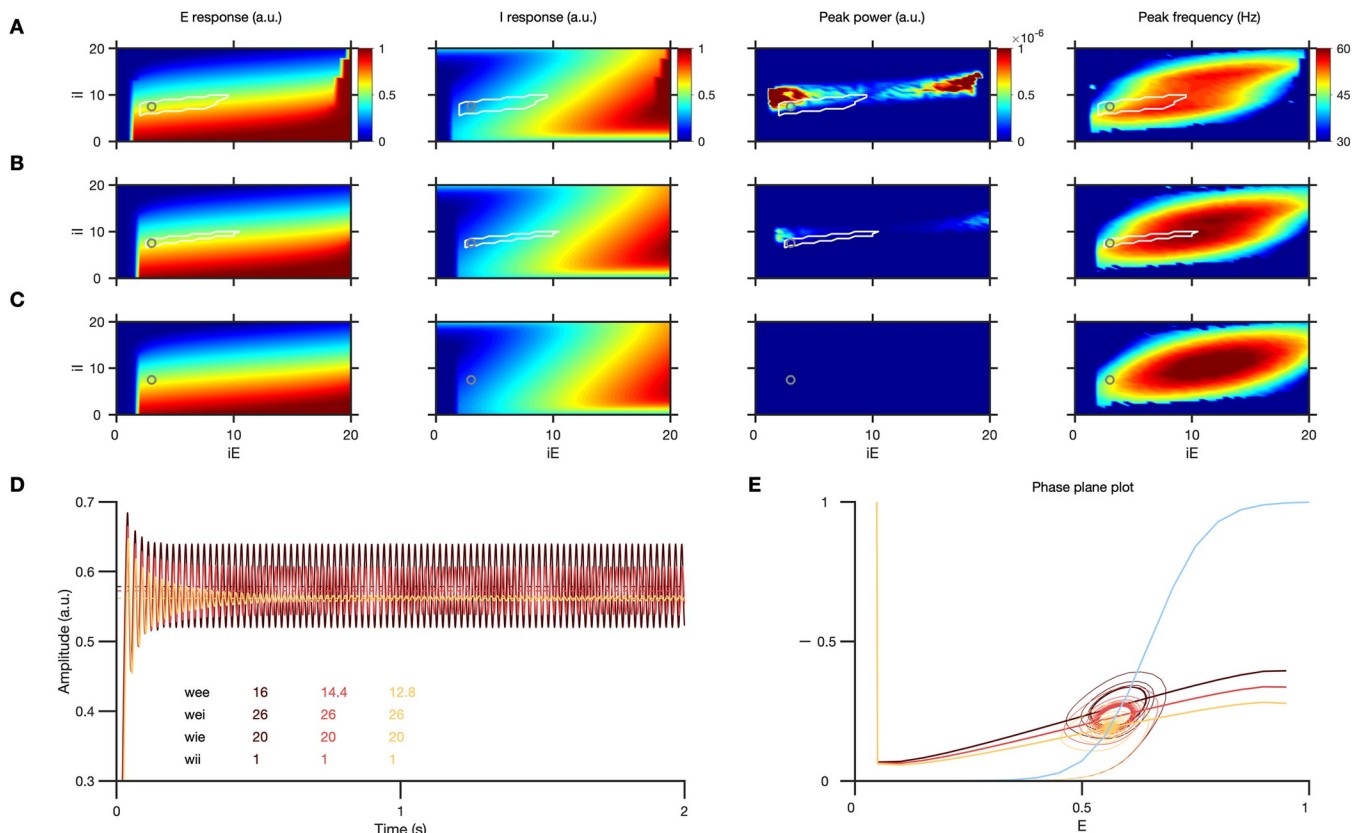

**Fig 8. Effect of stimulus discontinuity in the model.** (A) Left to right: E and I response, peak power in the gamma range and the corresponding peak frequency, for different values if *iE* and *iI* for the default model weights. (B) and (C) are the corresponding results for modified networks with only *wee* reduced by 10% and 20%, respectively, as excitatory LR inputs to E decrease due to discontinuity (weights indicated in (D)). White lines indicate the sublinear E and superlinear I response regime where increase in *iI* causes gamma amplitude to increase and frequency to decrease (this regime is absent in case 3). Oscillations weaken drastically in the network from case 1 to 2 and 3, even when the mean E and I responses do not change as much. (D) E response in the above 3 cases for *iE* = 3, *iI* = 7.5 (marked by the gray circle). (E) Phase-plane analysis showing the effect of varying *wee* on the E and I nullclines. Only the E nullcline shifts in these 3 cases, while I nullcline (light blue) remains the same. Figure data are located at https://doi.org/10.5281/zenodo.6523772. LR, lateral recurrent.

*iI*). Here, increasing the size of the stimulus increases *iI*, while increasing the contrast of the stimulus increases both *iE* and *iI*. Jadi and Sejnowski [14] showed that in an oscillatory regime in which the responses of I population is strongly superlinear while E population is sublinear (as indicated by white lines; for details see [14]), increasing stimulus contrast increases gamma peak frequency and increasing stimulus size decreases gamma peak frequency and increases its magnitude, as observed in real data [10,19,20,24,25].

Here, we show that small perturbations in the model operating within this regime, which only lead to small changes in the steady-state firing rates of the E and I populations, can severely attenuate gamma oscillations. We modeled the discontinuity as a drop in the excitatory synaptic weights for E (*wee*) by 10% (Fig 8B) and 20% (Fig 8C). Even a 10% drop in *wee* caused oscillations to weaken considerably (Fig 8B and 8D), and they were almost abolished with a 20% reduction (Fig 8C and 8D). Importantly, the mean E or I response remained almost unchanged, although there is a slight drop in this case, contrary to the experimental results where spiking responses mostly increase due to discontinuity. To study the properties of this dynamical system, we plotted the excitatory and inhibitory nullclines and found that reduction in *wee* led to a minor flattening of the excitatory nullcline (Fig 8E). While this led to only a minor change in the fixed point (mean E and I responses), the oscillatory power (limit cycle)

reduced drastically. Jadi and Sejnowski [14] had shown that by varying the external inputs (which cause a translation of the E and I nullclines), the dynamical system can transition from a stable operating regime to an oscillatory one through a transition known as a supercritical Andronov–Hopf bifurcation, in which both the power and frequency of the oscillations critically depend on the weights and time constants. Such a finely balanced system near the bifurcation point could be brought back to a stable regime by a slight reduction of the excitatory weights, leading to the loss of oscillations. Modifications in other weights (such as *wii*, *wei*, or *wie*), which caused similar changes in the slopes of the E and I nullclines, also led to changes in the oscillatory properties and the expanse of the valid regime where the model behaved as expected. Reduction in weights mostly led to shrinking of this valid regime, and concomitant changes in weights produced even stronger effects. Overall, these results demonstrate a resonance-like behavior of the oscillations, similar to the experimental observations, whereby small perturbations that do not change the steady-state firing rates by much can nonetheless cause huge changes in the oscillations.

## Discussion

Strong gamma oscillations in V1 LFP are known to be induced by optimized grating stimuli. When the center and surround of V1 RF were stimulated by gratings of different orientations, phases, or contrasts, or when there was a gray annular cut between these regions, strength of gamma oscillations was severely reduced. Importantly, gamma oscillations were more sensitive to these discontinuities compared to spiking activity. Furthermore, this sensitivity of gamma to the discontinuity was dependent on the location of discontinuity with respect to the center-surround structure of the V1 RF. We demonstrate that an E-I neuronal network model can exhibit such a behavior when there are small changes in overall weights in the network due to such discontinuities.

### Gamma and visual irregularities

Gamma oscillations induced by large gratings have been shown to be stronger and coherent over larger spatial extents than those induced by small gratings, but are adversely affected by addition of noise [10,27,28], or superimposition of cross orientation gratings to form plaids [21,29]. While addition of noise or an orthogonal orientation component modifies both the center and surround properties, our manipulations mostly change the stimulus outside the classical RF, bringing out the strong influence of this extraclassical modulation on gamma oscillations. This modulation can be both inhibitory and excitatory, and is mostly mediated through inputs from lateral intracortical connections, feedback from higher areas, and feedforward inputs [13,30–33]. In a similar context, a study in mouse V1 [12] showed that orthogonal orientation in the extraclassical surround reduces the power of a "context-dependent gamma" rhythm. The image-computable model described by Hermes and colleagues [7], in which gamma depends on variance across multiple orientation channels, would also predict a reduction in gamma rhythm due to an unmatched orientation in the extraclassical surround. The specific biophysical mechanism by which the variance model is implemented could potentially be tuned to produce a large reduction in gamma with a small discontinuity, although that remains beyond the scope of this study.

Across all the discontinuity types, an additional contrast component introduced by the sharp stimulus transition across the discontinuity edge may contribute to the excitation of heterogeneous groups of neurons preferring other orientations and spatial frequency in V1, thereby adding to the E-I imbalance. In this context, the orientation and phase discontinuities have been shown previously to cause an effect called brightness induction or enhancement

[34,35], under which the center appears perceptually brighter due to the orientation contrast or phase contrast. A previous study [34] reported that the firing rates for the units responding to the center grating increased only with orientation discontinuity but not with phase discontinuity. We found that firing rates increased for both the stimulus variations, as well as for the annular cut case, which has been shown to not cause any brightness enhancement, although the change was not as pronounced as the change in gamma power. The effect of stimulus manipulations was generally more diverse on firing rates than gamma (S2 Fig). This is unsurprising because individual neurons could be excited or inhibited by stimulation, slight mismatch between the central stimulus size and RF sizes could lead to overall suppression or facilitation of responses, and the effect of stimulating the surround itself could be suppressive or mildly facilitative (see [13] for a review). Gamma, on the other hand, depends on population dynamics of a much larger population of neurons and therefore the effect of discontinuity was more homogeneous across electrodes. This also explains why the effect of discontinuity on spiking versus firing rates did not show any relationship (S2 Fig). Nonetheless, we note that the overall effect of discontinuity on firing rates was also robust, albeit of a smaller magnitude as compared to gamma. In general, however, it has been shown that gamma and spiking activity may not necessarily show a definite relationship across stimulus variations [10,29]. One important consideration here is that the stimulus features such as orientation, spatial frequency, and size were chosen to optimize gamma oscillations for the continuous stimulus case. These features may not always be the preferred ones for the spiking responses, since their tuning is more diverse than that for gamma, which remains more consistent across sites [18].

The nature of modulatory inputs from the extraclassical surround region is heavily dependent on the stimulus contrast [13,26]. For example, the surround region can be suppressive at high contrast but can provide net excitation at a lower contrast, and the size of the excitatory RFs in V1 expands slightly at lower contrasts in comparison to the higher contrasts [36]. These contrast-dependent effects may modulate the E-I signals in different ways at different contrasts in our stimulus set, leading to slight differences in the effect of discontinuities at different contrasts (Fig 6). The effect of discontinuities is qualitatively similar even when they progressively remove areas of spatial contrast, as in annular discontinuities, modifying the overall stimulated cortical volume itself. While our results pertain to grating stimuli, gamma oscillations have been observed to be reduced by a mismatched surround or annulus for uniform surfaces as well [6]. These findings suggest that any visual structural modification that changes the critical level and nature of E-I drive in the network can potentially affect gamma. Additionally, it seems that the essential substrate or the recurrent E-I network that can replicate the resonant behavior of gamma is contained within the cortex. Therefore, as indicated previously [37], gamma in V1 could mainly have a cortical source and is not inherited from the lateral geniculate nucleus.

## Animal-specific differences

While all the major results shown here were consistent across monkeys, the magnitude of these effects were in general smaller in M2 as compared to M1. One of the factors in this could be that the RFs in M1 were at a larger visual eccentricity than in M2. RFs closer to the fovea are known to be smaller, as seen in M2 here, and the properties of RFs in terms of overall connectivity, gains, and center-surround spread and interactions may depend on visual eccentricity. Another notable difference is the presence of a prominent slow gamma rhythm in M2. The effects of discontinuity on this rhythm were qualitatively similar to fast gamma, although the drop was smaller. In particular, for annular discontinuities at varying inner radii, slow gamma reduced strongly even at farther cuts, which is expected if a spatially more widespread network

is involved in its generation [18]. Moreover, the effect of smallest contrast discontinuity was most drastic for slow gamma in the mid-contrast range, whereas for fast gamma it was in the high-contrast range (Fig 5). This effect incidentally follows the contrast tuning of these 2 gamma rhythms to large continuous gratings [18].

## Implications for natural image processing

Given the high sensitivity of gamma to small stimulus discontinuities that we show, most natural stimuli might not be able to generate strong gamma oscillations in V1 LFP, as reported by many studies [3,5,7,38]. Gamma is also strongly generated by uniform color surfaces [39], and therefore some colored natural stimuli may generate strong gamma when favorable uniform color patches fall on the RF [3,4,38]. However, mismatches in color surfaces in the form of a differently colored blob or annulus can also reduce gamma power [6], suggesting that the effect of discontinuities is qualitatively consistent across both achromatic and chromatic pathways, and that most natural stimuli with intricate structural complexities would not generate strong gamma oscillations. Thus, it seems unlikely for gamma in V1 LFP to be playing a definitive role in the process of perceptual binding, although our stimuli and behavioral experiment are not set up ideally to directly test this.

Responses in V1 for naturalistic stimuli can be explained by the degree of statistical dependence between the stimulus structures falling on the center and surround regions of the V1 RF [40]. Considering the high sensitivity of gamma to this visual structure, it will be interesting to determine how gamma depends on natural scene statistics. Recently, it has been shown that information about images is high in the LFP gamma band [38], and orientation variability in images can be used to predict gamma responses [7]. Previous work has shown that extraclassical surround effects in V1 due to natural images can emerge from predictive feedback signals from higher to lower cortical areas [41]. In this context, it has also been suggested that gamma and firing rates play complimentary roles as predictive signals for lower and higher order image features respectively [42]. However, it is not clear how image predictability varies across the discontinuities used in our experiments, and whether it aligns with the observed nonlinear effects on gamma and spiking responses. It may be argued that gamma in V1 signifies the stimuli more optimized to V1 response properties and as an extension of this, gamma in higher areas might code for larger surfaces or whole objects more relevant to the response properties of those neurons. However, gamma strength may decrease with increasing eccentricity and RF size leading to less prominent oscillations in higher areas [43].

## Gamma mechanisms and neural models

Neuronal models with recurrent connections between excitatory and inhibitory populations have been used to model and study oscillations in cortical and subcortical networks [10,14–16,44,45]. Mechanisms of gamma may be understood better by studying how discontinuities affect the overall network dynamics in such models. Cortical neurons are known to have supralinear input–output response functions such that the net synaptic throughput effectively changes with input strength [46–48]. As the network gets engaged more strongly, as in the case of large gratings, normalization signals dominate and inhibition stabilizes the network activity [49–51]. The tuned nature of these signals can modulate network interactions depending on stimulus properties [13,45,52,53]. Therefore, discontinuities can modulate the overall network drive, which can effectively modify synaptic gains and the operating regime of the network. Incorporation of the detailed RF structure into the model may further unravel mechanistic properties of these oscillations. Recently, Heeger and Zemlianova [45] developed a family of models with recurrent amplification to implement normalization in a V1-like neuronal

network. Strong gamma oscillations are generated in some models of their family for a restricted set of stimulus conditions, and they depend on the strength of the normalization pool, which is also tuned. Exploration of such models under the light of our experimental findings could lead to a better understanding of how discontinuities affect spiking and oscillatory responses.

In summary, gamma oscillations in V1 LFP exhibit resonance-like behavior that could signify critical E-I balance in the neuronal network. Further investigations with finer variations of stimulus properties across this specialized RF structure may lead to better insights into the dynamics and mechanisms of such oscillations. Studying such oscillations in conjunction with spikes and their sensitivity to the spatiotemporal sensory context may help us understand general principles of cortical sensory processing.

## Materials and methods

### Animal preparation and training

All experiments were performed as per the guidelines approved by Institutional Animal Ethics Committee (IAEC) of the Indian Institute of Science (CAF/Ethics/239/2011) and the Committee for the Purpose of Control and Supervision of Experiments on Animals (CPCSEA, F.No. 25/27/2013-AWD/42.8). For this study, 2 adult female monkeys (*Macaca radiata*; 13 years, approximately 3.3 kg; 17 years, approximately 4 kg) were used. Animal preparation and training details are the same as described in earlier studies [18,39]. Each monkey was surgically implanted with a titanium headpost over the anterior/frontal region of the skull under general anesthesia. Following recovery, the monkey was trained on a passive visual fixation task, and once the monkey reached a satisfactorily level of training, another surgery was performed under general anesthesia to insert a microelectrode array (Utah array, 96 active platinum microelectrodes, 1 mm length, 400 μm inter-electrode distance, Blackrock Microsystems, Salt Lake City, UT, USA) in the primary visual cortex (right hemisphere, centered at approximately 10 to 15 mm rostral from the occipital ridge and approximately 10 to 15 mm lateral from the midline; the location varied slightly across the 2 monkeys). The RFs of the recorded neurons were located at an eccentricity of between approximately 3˚ to approximately 4.5˚ in M1 and between approximately 1.4˚ to approximately 1.8˚ in M2, in the lower left quadrant of the visual space with respect to fixation (S1 Fig). Considering the dimensions of the microelectrodes, the recordings are most likely to be around cortical layer 2/3.

### Experimental setup and behavior

Details of experimental setup, behavior, and data recording are as described in previous studies [18,39]. Each monkey viewed a monitor (BenQ XL2411, LCD, 1,280 × 720 resolution, 100 Hz refresh rate, gamma corrected and calibrated to a mean luminance of 60 cd/m$^2$ on the monitor surface using i1Display Pro, x-rite PANTONE) placed approximately 50 cm from its eyes, with its head fixed by the headpost in a custom designed monkey chair. A Faraday enclosure (constructed using thin copper sheets, wood, and sound-isolating material), with a dedicated grounding separate from the mains supply ground, was used to house the monkey chair and the display and recording setup during experiments.

The monkeys performed a passive visual fixation task, in which they had to maintain visual fixation at a small dot (0.05˚ or 0.10˚ radius) at the center of the screen for the duration of a trial, which could be either 3.3 or 4.8 s. Each trial began when the monkey fixated, and following an initial blank gray screen of 1,000 ms, 2 to 3 stimuli were shown for 800 ms each, with an inter-stimulus interval of 700 ms. The monkey was rewarded with juice for successfully holding its fixation without blinking, within 2˚ of the fixation spot, which stayed on throughout

this period. Although the fixation window was kept slightly large, mainly to adjust for occasional small shifts in the head position due to slight movements of the chair and related apparatus, the monkeys actually maintained fixation within a much smaller window during the task. The standard deviation of eye position during a trial across sessions was small on an average for both monkeys ($<0.18°$ and $<0.16°$ along the horizontal and vertical axes, respectively, for M1; $<0.28°$ and $<0.29°$ for M2).

## Data recording

A 128-channel Cerebus Neural Signal Processor (Blackrock Microsystems) was used to record raw signals on 96 channels. To obtain the LFPs, these raw signals were filtered online between 0.3 Hz (Butterworth filter, first order, analog) and 500 Hz (Butterworth filter, fourth order, digital), sampled at 2 KHz and digitized at 16-bit resolution. To extract multiunit spikes, the raw signals were filtered online separately between 250 Hz (Butterworth filter, fourth order, digital) and 7.5 KHz (Butterworth filter, third order, analog), and the filtered signal was subjected to a threshold (amplitude threshold of approximately 5 standard deviations of the signal). No further offline filtering of the LFP signals or offline spike sorting was done.

An ETL-200 Primate Eye Tracking System (ISCAN Incorporated, Woburn, MA, USA) was used to record eye position data in terms of horizontal and vertical co-ordinates/position and pupil diameter, at a sampling rate of 200 Hz during the task. A custom software running on MAC OS monitored the eye signals, and controlled the progression of task and trials, stimulus generation and pseudorandom stimulus presentation.

## Electrode selection and data analysis

For each monkey, an RF mapping experiment was run regularly across multiple sessions across days to verify the stability of RFs and assess the suitability of electrodes for data analyses. In this experiment, small sinusoidal gratings (radius of 0.3° and 0.2° for M1 and M2, respectively; static, full contrast, spatial frequency of 4 cpd, at 4 orientations of 0°, 45°, 90°, and 135° in both monkeys) were flashed for 200 ms at equally spaced ($9 \times 9$) locations within a rectangular grid on the visual space that approximately covered the aggregate RF of the entire microelectrode array. When such stimuli were presented near the RF, they produced a negative deflection in the evoked response between 40 to 100 ms after stimulus onset (see Fig 2A of [54] for the evoked responses). We computed the "response" as minimum LFP value between 40 to 100 ms, and further "baseline corrected" it by subtracting the response computed between –100 to –40 ms of stimulus onset. A 2D Gaussian fitted to the spatial profile of this response induced by stimuli presented at the various locations yielded the RF estimate for each electrode [54]. Electrodes with consistent stimulus induced changes in LFP and reliable estimates of RF size across sessions were chosen after subjecting their mean distribution across sessions to an arbitrary threshold based on inspection (S1B Fig). Since we primarily analyzed and characterized LFP responses, an LFP-based measure was used, although estimates based on spiking activity were similar (S1E and S1F Fig) [17,54]. Electrodes that showed noisy or inconsistent signals, or a high degree of crosstalk across sessions, or impedances outside the range 250 KΩ to 2,500 KΩ, were discarded from analyses. This procedure yielded an overall 65 and 39 usable electrodes for M1 and M2, respectively, which were considered for all further analyses (see S1 Fig for the estimated RFs and related details). Out of these, the electrodes that showed high impedance during certain sessions were not considered for analysis for that session. The resultant set of electrodes were considered while choosing the center sites/electrodes (explained ahead) during any session. The final number of electrodes used for analysis of data from different sessions is stated in the figures or in the related text descriptions.

For each session, the sites whose RFs were sufficiently close to the stimulus center (within a range of 0.2˚ and 0.15˚ for M1 and M2, respectively; a smaller range was chosen for M2 since the RFs were less eccentric and smaller than in M1) were selected as the "center" sites for analyses. For the annular cut discontinuity experiment with different cut locations, this range was 0.15˚ and 0.1˚ for M1 and M2, respectively, corresponding to the nearest cut location. All analyses were done for such sites, and the mean was taken across sites across sessions.

Every experiment was repeated over several sessions, each yielding data from a few sites whose RFs were close to the center of the stimulus. After rejection of electrical artifacts or noisy data (average rejection rate: 1.57% in M1, 2.07% in M2; procedure summarized in [18]), the number of sessions, number of center electrodes, and the mean number of repeats across sessions, for the 2 monkeys, respectively, were as follows: Annular cut discontinuity experiment (Fig 1): 13 sessions, 60 electrodes (out of which 44 were unique, since some electrodes were repeated across sessions), 11.02 repeats for M1 and 8 sessions, 53 (32 unique) electrodes, 15.16 repeats for M2; Annular cut discontinuity experiment with different cut locations (Figs 2 and S1): 13 sessions, 33 (25 unique) electrodes, 11.93 repeats for M1 and 7 sessions, 21 (17 unique) electrodes, 15.23 repeats for M2; Orientation discontinuity experiment (Fig 3): 11 sessions, 57 electrodes (43 unique) and 12.32 repeats for M1 and 9 sessions, 74 electrodes (32 unique) with 14.55 repeats for M2; Phase discontinuity experiment (Fig 4): 12 sessions, 62 (44 unique) electrodes, 12.42 repeats for M1 and 13 sessions, 124 (37 unique) electrodes, 16.09 repeats for M2; Contrast discontinuity experiment (Figs 5 and 6): 9 sessions, 48 (42 unique) electrodes, 11.95 repeats for M1 and 6 sessions, 48 (27 unique) electrodes, 14.72 repeats for M2. All these data analyses were performed using custom written codes in MATLAB, MathWorks.

## Spectral analyses

Spectral analyses were performed using the Multitaper method, implemented using functions in the "Chronux toolbox" [55] (http://chronux.org/, developed for MATLAB). To obtain the time-frequency difference spectrum, the power spectrum was first calculated using a single taper with a sliding window of 0.25 s yielding a 4-Hz frequency resolution. The logarithm of the mean power spectrum across repeats was computed for each electrode. Then, the mean power at each frequency in the mean spectrum during the spontaneous period (0.5 to 0 s before stimulus onset) was calculated and subtracted from the entire spectrum, followed by multiplication by 10 to get the difference spectrum in decibel.

Power spectral density (PSD) were computed using a single taper between 0.25 to 0.75 s after stimulus onset and compared to the PSD during spontaneous period (0.5 to 0 s). The change in power was calculated by subtracting the logarithm of power during stimulus period and spontaneous activity and multiplying by 10 to yield units in decibels.

Power in any frequency band is calculated as the sum of PSD at all frequencies in that band. Normalized gamma power was calculated for each electrode by first calculating the mean gamma power for each condition and then dividing these by their maximum value across conditions for that electrode.

Large gratings that substantially extend into the surround region of the RF can induce 2 distinct gamma rhythms (slow and fast), whose strength and peak frequency can vary with the grating orientation [18]. Since the stimuli in this study were optimized to generate strong fast gamma rhythms, analyses and results are focused on this band. Therefore, across all figures and results "gamma" refers to fast gamma unless stated otherwise. For all static grating stimuli at 100% contrast (Figs 1–4), fast gamma band was chosen as per inspection of PSD as: [35 65] Hz in M1 and [45 75] Hz in M2. The same bands were chosen in the contrast discontinuity

experiment, for stimuli in which at least one of the inner or the outer grating was at 100% contrast. For others, in which both inner and outer grating were at a contrast lower than 100%, the bands were shifted lower by 5 Hz, since gamma rhythm is known to shift lower in frequency with decreasing contrast, and as also observed in the corresponding PSD.

## Analyses of spiking units

The center units that showed clear spikes and had an average firing rate of at least 1 spike/s for at least one of the stimulus conditions were chosen for analyses of firing responses. The firing rates for each unit were normalized by dividing by the maximum firing rate for that unit across time and conditions to obtain the mean normalized firing rate for each condition (Figs 1C, 3C, 4C, 5C and S1C). For comparison of normalized gamma power and normalized firing rate, these same spiking units were considered. The stimulus period mean normalized value of gamma power and firing rate for each condition were calculated for each selected spiking center electrode by first calculating the corresponding average value during the stimulus period (0.25 to 0.75 s after stimulus onset), followed by normalization by the maximum value across conditions, and averaging these quantities across these chosen center electrodes across sessions (Figs 1D, 3D, 4D, 6A and 6B).

## Model details

The model consists of 2 interconnected neuronal populations representing a local recurrent network in an orientation column of V1, with external inputs to the network arising from stimulation and lateral or feedback signals. If E and I are considered as the response variables representing the mean response of the 2 neuronal populations, then the equations governing the dynamics of E and I are given by,

$$\tau_x \frac{dX}{dt} = -X + f_x(I_x).\tag{1}$$

Where $x$ can be E or I. $\tau_x$ is the corresponding time constant of the variable build up, and $f_x$ is the function transforming the total synaptic input, $I_x$, to the resultant spiking response.

$$I_e = W_{ee}E - W_{ei}I + iLR_e + iE \tag{2}$$

$$I_i = W_{ie}E - W_{ii}I + iLR_i + iI \tag{3}$$

$f_x$ is considered to be a sigmoid function as described in [14].

$$f_x(I_x) = \frac{1}{1 + e^{-m_x(I_x - \theta_x)}} - \frac{1}{1 + e^{m_x \theta_x}} \tag{4}$$

The external inputs arising from the center (inner) and surround (outer) stimuli are $iE$ and $iI$ for E and I, respectively. These equations are similar to the model used by Jadi and Sejnowski [14], except there were no separate lateral inputs ($iLR_e$ and $iLR_i$) in their model. We added $iLR_e$ and $iLR_i$ as the effective intracortical inputs from similar nearby local recurrent input networks through interconnections in the orientation hypercolumn [13,53]. $W_{xy}$ implies the synaptic gain from input $y$ to the receptor $x$. Considering that all nearby local recurrent networks differ only in their orientation tuning, $iLR_x$ is effectively a weighted function of the E and I activity, for the sake of this simplified single-column model design. Therefore, Eqs 2 and 3 can be rewritten as follows.

$$I_e = W_{ee}E - W_{ei}I + W_{eELR}E - W_{eILR}I + iE \tag{5}$$

$$I_i = W_{ie}E - W_{ii}I + W_{iELR}E - W_{iILR}I + iI \qquad 6$$

$W_{xELR}$ and $W_{xILR}$ denote the gain function for the excitatory and inhibitory inputs, respectively, from the other local recurrent networks in the hypercolumn. Note that in a multi-column architecture, these inputs ($iLR_x$) can be derived from the actual outputs of the nearby columns. Eqs 5 and 6 can be written as follows.

$$I_e = (W_{ee} + W_{eELR}) E - (W_{ei} + W_{eILR}) I + iE \qquad 7$$

$$I_i = (W_{ie} + W_{iELR}) E - (W_{ii} + W_{iILR}) I + iI \qquad 8$$

In this model, discontinuities can effectively decrease $W_{xELR}$ and/or $W_{xILR}$, and thereby the overall network gains, pushing the network into a different regime of operation.

Eqs 7 and 8 can be summarized as follows.

$$I_e = w_{ee}E - w_{ei}I + iE \qquad 9$$

$$I_i = w_{ie}E - w_{ii}I + iI \qquad 10$$

$$w_{ee} = W_{ee} + W_{eELR}, w_{ei} = W_{ei} + W_{eILR}, w_{ie} = W_{ie} + W_{iELR}, w_{ii} = W_{ii} + W_{iILR} \qquad (11)$$

The default values of parameters, adapted from [14], are as follows. $wee = 16$, $wei = 26$, $wie = 20$, $wii = 1$, $m_E = 1$, $m_I = 1$, $\theta_E = 5$, $\theta_I = 20$, $\tau_E = 20$, and $\tau_I = 10$. These parametric values have been shown to generate gamma oscillations in the network whose properties (peak power and frequency) agree with experimental observations under certain regimes of the network operation, and the network behaves as an inhibition stabilized network (ISN) [14,51]. In Fig 8, only $W_{eELR}$ is considered to be reduced by a discontinuity such that ($W_{ee} + W_{eELR}$), i.e., $wee$, effectively decreases by 10% and 20%, and the network moves drastically to a regime with weak oscillations under new weights. To determine the regime where increase in $iI$ caused an increase in the magnitude of oscillation and decrease in its peak frequency, we followed the conditions derived in Jadi and Sejnowski [14] (Appendix B, Eqs 6, 7, 12, 13 and 14) and the procedure described therein. The exact dependence of peak power and frequency of the oscillation on the network weights and time constants, within the oscillatory regime, is derived in Jadi and Sejnowski [14] and is therefore not replicated here.

## Supporting information

**S1 Fig. RF mapping.** (A) Magnitude of the evoked response produced by mapping stimuli (for details, see Materials and methods) is shown across the grid of electrodes, averaged across RF mapping sessions (7 sessions each in M1 and M2). (B) Electrodes showing a response above an arbitrary threshold (100 for M1, 60 for M2) are chosen as consistently good electrodes (indicated in red) for further analyses. (C) A color-coded schematic of the physical microelectrode grid and (D) the corresponding RF centers of these sites in the visual space determined as per the mapping algorithm described in the Materials and methods. Mapping based on firing rates (MUA) was done similarly by using the change in mean firing rate from baseline to stimulus period as the "response" metric. (E) Scatter of the estimated azimuth and elevation of RF centers for the selected electrodes. The RF center estimated from LFP and MUA were closely matched in both monkeys. (F) Scatter of RF size across the selected electrodes. Mean sizes are shown in red. Figure data are located at https://doi.org/10.5281/zenodo.6523772. LFP, local field potential; M1, Monkey 1; M2, Monkey 2; MUA, multiunit activity;

RF, receptive field.
(PDF)

**S2 Fig. Slope of regression of gamma and firing rate with spatial, orientation and phase discontinuity.** Slope of regression with (A) annulus width, (B) orientation discontinuity ($O$-$I$)°, and (C) phase discontinuity ($O$-$I$)$\phi$° (Δvalue/degree of visual angle) of normalized gamma power versus normalized firing rate. The format of this figure is the same as Figs 1E, 3E, and 4E. The slope of regression for gamma was consistently negative across sites in both monkeys for every type of stimulus discontinuity. The corresponding effects for firing rates were more heterogenous. Therefore, to compare the sensitivity of gamma and firing rates to different stimulus discontinuities, we used the magnitude of slope of regression (Figs 1E, 3E, and 4E). We found no relationship between the slopes for gamma and firing rates (linear regression of slopes for gamma on slopes for firing rates: spatial discontinuity in (A) $R^2$ = 0.002, $p$ = 0.78 in M1, $R^2$ = 0.16, $p$ = 0.01 in M2; orientation discontinuity in (B) $R^2$ = 0.003, $p$ = 0.76 in M1, $R^2$ = 0.03, $p$ = 0.22 in M2; and phase discontinuity in (C) $R^2$ = 0.000, $p$ = 0.97 in M1, $R^2$ = 0.02, $p$ = 0.27 in M2). Figure data are located at https://doi.org/10.5281/zenodo.6523772. M1, Monkey 1; M2, Monkey 2.
(PDF)

**S3 Fig. Effect of annular discontinuity at different locations.** (A) Trial-averaged time-frequency difference spectra for the population in M1 (left column) and M2 (right column), induced by stimuli with annular discontinuity at different inner radii (inner radius and annulus width are indicated on the left and top, respectively) and (B) the corresponding mean change in power from baseline to stimulus period. (C) Mean normalized firing rate averaged across selected spiking center electrodes and sessions. Figure data are located at https://doi.org/10.5281/zenodo.6523772. M1, Monkey 1; M2, Monkey 2.
(PDF)

## Author Contributions

**Conceptualization:** Vinay Shirhatti, Supratim Ray.

**Data curation:** Vinay Shirhatti, Poojya Ravishankar.

**Formal analysis:** Vinay Shirhatti.

**Funding acquisition:** Supratim Ray.

**Investigation:** Vinay Shirhatti, Poojya Ravishankar, Supratim Ray.

**Methodology:** Vinay Shirhatti, Poojya Ravishankar, Supratim Ray.

**Project administration:** Supratim Ray.

**Resources:** Supratim Ray.

**Software:** Vinay Shirhatti, Supratim Ray.

**Supervision:** Supratim Ray.

**Validation:** Vinay Shirhatti, Supratim Ray.

**Visualization:** Vinay Shirhatti, Supratim Ray.

**Writing – original draft:** Vinay Shirhatti.

**Writing – review & editing:** Vinay Shirhatti, Supratim Ray.

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
