## [Editor Report · Decision Letter 0]

14 Jan 2022

Dear Dr Ray, 

Thank you for submitting your manuscript entitled "Gamma oscillations in primate primary visual cortex are severely attenuated by small stimulus discontinuities" for consideration as a Research Article by PLOS Biology.

Your manuscript has now been evaluated by the PLOS Biology editorial staff, as well as by an academic editor with relevant expertise, and I am writing to let you know that we would like to send your submission out for external peer review.

Once your full submission is complete, your paper will undergo a series of checks in preparation for peer review. Once your manuscript has passed the checks it will be sent out for review. To provide the metadata for your submission, please Login to Editorial Manager (https://www.editorialmanager.com/pbiology) within two working days, i.e. by Jan 18 2022 11:59PM.

If your manuscript has been previously reviewed at another journal, PLOS Biology is willing to work with those reviews in order to avoid re-starting the process. Submission of the previous reviews is entirely optional and our ability to use them effectively will depend on the willingness of the previous journal to confirm the content of the reports and share the reviewer identities. Please note that we reserve the right to invite additional reviewers if we consider that additional/independent reviewers are needed, although we aim to avoid this as far as possible. In our experience, working with previous reviews does save time. 

If you would like to send previous reviewer reports to us, please email me at ggasque@plos.org to let me know, including the name of the previous journal and the manuscript ID the study was given, as well as attaching a point-by-point response to reviewers that details how you have or plan to address the reviewers' concerns. 

Given the disruptions resulting from the ongoing COVID-19 pandemic, please expect some delays in the editorial process. We apologise in advance for any inconvenience caused and will do our best to minimize impact as far as possible.

Kind regards,

Gabriel

Gabriel Gasque

Senior Editor

PLOS Biology

ggasque@plos.org

---

## [Decision Letter · Decision Letter 1]

14 Mar 2022

Dear Dr Ray,

Thank you for submitting your manuscript entitled "Gamma oscillations in primate primary visual cortex are severely attenuated by small stimulus discontinuities" for consideration as a Research Article at PLOS Biology. Thank you also for your patience as we completed our editorial process, and please accept my apologies for the delay in providing you with our decision. Your manuscript has been evaluated by the PLOS Biology editors, an Academic Editor with relevant expertise, and by three independent reviewers.

As you will see, the reviewers find the conclusions of the manuscript novel and interesting, but they also raise several issues that would need to be addressed. In light of the reviews (attached below), we are pleased to offer you the opportunity to address the comments from the reviewers in a revised version that we anticipate should not take you very long. We will then assess your revised manuscript and your response to the reviewers' comments and we may consult the reviewers again.

In addition, we would like to suggest a title that is more informative and appealing to a broad readership:

“Gamma oscillations in primary visual cortex are a signature of excitation-inhibition balance and attenuated by small stimulus discontinuities.”

However, we would be happy to work with you on an alternative if you think our suggestion misrepresents your findings. 

We expect to receive your revised manuscript within 1 month.

**IMPORTANT - SUBMITTING YOUR REVISION**

3. Resubmission Checklist

a) *Ethics Statement*

Thank you for including the ethics statement. Please include the license/approval number of your experimental protocols. 

b) *PLOS Data Policy*

Please provide the data underlying the graphs shown in the following figures:

Figures 1A-E, 2A-C, 3A-E, 4A-E, 5A-C, 6A-C, 8A-E, and S1A-C

Please also ensure that each figure legend in your manuscript includes information on WHERE THE UNDERLYING DATA CAN BE FOUND and that your supplemental data file/s has/have a legend.

**Please note that all the data will have to be publicly available to be able to accept the manuscript for publication.

c) *Published Peer Review*

Sincerely,

Ines

--

Ines Alvarez-Garcia, PhD

Senior Editor

PLOS Biology

Reviewers' comments

Rev. 1:

The authors recorded from V1 neurons/locations in passively viewing monkeys, using chronically implanted Utah arrays. They varied stimulus discontinuities parametrically along 4 feature dimensions: annular cut discontinuity, orientation discontinuity, phase discontinuity and contrast discontinuity to study the effects of discontinuities on LFP (high/fast) gamma power, and on spiking activity. They report that for all discontinuities tested gamma power is reduced rapidly, even for small discontinuities, and the changes in gamma power are more pronounced than changes in spiking activity. They build a computational model to explain these effects and report that small changes in excitatory weights (although changes in other weights also affected oscillatory behaviour) in the model result in radically changes in oscillatory behaviour, with only limited effects on overall firing rates. The authors hence argue that changes in gamma power in V1 with stimulus discontinuities is a reflection of E-I balance in the network.

This is an interesting extension of previous work, and adds to an important discussion/controversy in the field. While the authors set the framework at least partly in light of the 'binding by synchrony', debate, they do not mention it in the discussion? Maybe this is because the stimuli used are perceptually 'unbound', so they may not provide the ultimate test? 

Minor:

* line 59: full stop missing

* the authors might consider referencing Bartolo et al.,2011, EJN, who reported related findings. 

* line 315: 'in both the monkeys ..' change to 'in both monkeys'

* p value reporting: is it meaningful to report exact p-values that are <0.001? It makes more sense to report t-values, z-values, F-values and effect sizes. 

Rev. 2:

In this work Shirhatti et al. record from macaque primary visual cortex, quantifying changes in spiking and LFP gamma activity to grating stimuli with center-surround discontinuities. The authors show that specifically LFP gamma activity is strongly reduced with progressive stimulus discontinuities across several feature domains. These results are a critical and timely test of growing evidence to suggest visual gamma oscillations are highly sensitive to the structure of visual inputs, which is of functional significance. The authors interpret their findings within the context of excitatory-inhibitory dynamics with visual circuits, providing a computational model based on this interpretation which recapitulates several experimental observations. Overall, these findings are novel and important for understanding the mechanisms of gamma and its function. Below I note several areas where the authors findings require additional discussion.

Main comments:

-Introduction, paragraph (line 53). The authors contrast two alternative accounts for considering the effects of stimulus discontinuities on gamma. However, I think this framing should be reconsidered. Hermes et al. propose a forward encoding model of how stimulus features can be predictive of the observed strength in gamma responses. It is not a biophysical model, and is therefore compatible with several physiological implementations. It is therefore not incompatible with the next mentioned E-I balance models, which focus on circuit mechanisms and not stimulus encoding models. Given that both accounts predict reduced gamma as stimulus discontinuity increases, it seems more accurate to frame these approaches as describing the same putative phenomena at different levels, for which the current study provides critical data.

-Related to the above point, it would therefore be very compelling if in addition to the biophysical model presented, the authors also employed the Hermes et al. stimulus OV model on their stimuli/data. However, I appreciate that this might be beyond the scope of the manuscript. 

-Accurate receptive field (RF) mapping is critical to the design and interpretation of the authors findings. It therefore seems essential that the RF mapping data be presented as an initial figure. The authors should present RF stimulus/response maps with associated RF fitting for both monkeys. Secondly, in comparing spiking and LFP data, is there a difference in the measurement RF that must be considered (e.g. effectively larger RFs for LFP signals)?

-For the first experiment, the authors focus on the nature of regression slopes of spiking and gamma power changes across conditions, but fail to specifically note that these are in opposing directions. In general, the clear dissociation (often anticorrelation) between spiking and gamma effects seems less noted. However, this divergence in measures seems important for understanding the mechanisms and functional significance of these findings.

-Related to the comment above, many results show a clear anti-correlation or reciprocal relationship between population spiking and gamma results. However, individual scatter plots suggest that this relationship is weak (cf. Fig 1 D vs E). In the discussion, the authors should suggest why this might be. While also noting the implications for why conditions which favor gamma are, on average, associated with low/suppressed spiking.

-Several experiments show not only a reduction in gamma power with increasing discontinuities, but also an increase in the peak frequency. Is this a consistent phenomenon and how might it fit with the authors E-I mechanism?

-Several studies in primates report no evidence of gamma oscillations in the thalamus (LGN), suggesting a cortical mechanism. It appears the authors interpretation support this view, given the focus on E-I physiology and connectivity within visual cortex. This should be noted in the discussion.

-How do the authors experimental and computational findings relate to the similar modelling work of Heeger and Zemlianova (2020, PNAS)? (which should be cited).

-Overall, recent experimental and theoretical work has repeatedly linked center-surround effects in V1 to the genesis of strong gamma oscillations. Importantly, for gamma, it is congruency between center-surround stimulus features which is optimal. Alternatively put, classic spiking surround-suppression conditions (which decreases spiking) enhance gamma. Given these tight links to a larger literature, and recent arguments about such a mechanism reflecting a 'predictive' function of gamma, it would be important for the authors to directly engage how their findings/interpretation relate to this literature. Specifically, how does the authors E-I account align with the known physiology of surround-suppression, such that it's consistent with spike reduction and gamma enhancement. While I note the authors do engage with these ideas, the implication is often just that the E-I balance will be changed- which verges on being trivial (true for any input/event in cortex). Given the consistent parametric effects on gamma observed, should their not be a more specific interpretation for how the E-I balance is specifically changing under these conditions? 

Minor comments:

-While experimental conditions are color matched via stimuli shown in figures, legends for lower panels where color line plots are used would benefit the reader.

Rev. 3:

This study measures two kinds of neural signals, gamma oscillations and action potentials, from monkey primary visual cortex using microelectrode arrays. The main question addressed is how discontinuities in visual stimuli affect the two signals. Several discontinuities are introduced between the center and surrounding regions of an image patch, approximately centered over the receptive field of the mircroelectrodes. The discontinuities include a zero-contrast annulus, an orientation difference, a phase difference, and a contrast difference. In all 4 cases, the extent of the discontinuity is systematically varied (width of the annulus, orientation difference, phase difference, contrast difference). The main finding is that all discontinuities tend to result in a large decrease in the power of gamma oscillations and a smaller increase in the spike rates. The motivation for the study is not entirely clear, but generally it is suspected that image discontinuities are likely to impact gamma oscillations, and that this might provide some clues into the functional and biological significance of the oscillations. Finally, a widely used neural model comprised of interconnected excitatory and inhibitory units ("E/I") is implemented. The model does not directly allow for stimulus properties to be input, but discontinuities in the stimuli are simulated by varying some model parameters, which in turn shows that small perturbations in model parameters can result in large changes in oscillatory behavior of the model, with little change in mean signal level, interpreted as supporting the empirical observations. The main contribution of the paper is the empirical findings from the systematic manipulation of multiple types of stimulus discontinuities while measuring LFP and spiking. The fact that relatively small image discontinuities have large effects on the oscillations is likely to be important for understanding the significance (and perhaps biological origin) of the signals, though no strong conclusions are drawn about this. Nonetheless, the results are clear and likely to be of interest to the field. 

Major comments:

(1) I see no major problems with the data analysis or experimental methods. But surprisingly, what seems to me to be the biggest and most consistent effect in the dataset is barely commented on at all: the fact that discontinuities in the stimuli cause firing rates to increase and oscillations to decrease. The fact that the two signals are affected in opposite ways might have some implications for neural coding, and it is odd that the authors are primarily only interested in comparing the absolute value of the two signals, rather than the fact they change in opposite directions as a function of their stimulus manipulations.

(2) The firing rates are discussed as though they are only slightly modified by the discontinuities, compared to the enormous effects on oscillations. Yet the firing rate effects look robust as well. Clearly the effects on oscillations are bigger, but the firing rate effects are not small: they look to be about half the size of the effect on oscillations. Without a theory to help us interpret the meaning of these effects, it is difficult to say whether one, both, or neither, are big enough to matter. In any case, the difference between the two is somewhat overstated in places. 

(3) It looks like in the model, the effects on firing rates are smaller and go in the opposite direction compared to the data (if I am understanding the results correctly). This does not take away from the fact that the model shows much larger sensitivity of the oscillations than of the mean with respect to model parameters. But it would seem to warrant mention.

Overall, this is a very well done empirical study.

---

## [Editor Report · Decision Letter 2]

4 May 2022

Dear Dr Ray,

Thank you for submitting your revised Research Article entitled "Gamma oscillations in primate primary visual cortex are severely attenuated by small stimulus discontinuities and could be a signature of excitation-inhibition balance" for publication in PLOS Biology. I have now obtained advice from the Academic Editor and the rest of team of editors. 

Based on the discussion, we will probably accept this manuscript for publication, provided you satisfactorily address the data and other policy-related requests stated below.

In addition, we have discussed again the title and we have decided that the original version was the best one, thus please change it back to:

"Gamma oscillations in primate primary visual cortex are severely attenuated by small stimulus discontinuities"

We expect to receive your revised manuscript within two weeks. 

*Published Peer Review History*

*Press*

Sincerely,

Ines

--

Ines Alvarez-Garcia, PhD

Senior Editor

PLOS Biology

ETHICS STATEMENT:

-- Please include approval number of your experimental protocols.

DATA POLICY:

Thank you for including the data. We cannot accept sole deposition of data or code to GitHub (https://journals.plos.org/plosbiology/s/data-availability). However, you can archive this version of your publicly available GitHub data to Zenodo. Once you do this, it will also generate a DOI number that you can provide us with. See the process for doing this here: https://docs.github.com/en/repositories/archiving-a-github-repository/referencing-and-citing-content

---

## [Editor Report · Decision Letter 3]

10 May 2022

Dear Dr Ray,

On behalf of my colleagues and the Academic Editor, Christopher Pack, I am delighted to say that we can in principle accept your Research Article entitled "Gamma oscillations in primate primary visual cortex are severely attenuated by small stimulus discontinuities" for publication in PLOS Biology, provided you address any remaining formatting and reporting issues. These will be detailed in an email that will follow this letter and that you will usually receive within 2-3 business days, during which time no action is required from you. Please note that we will not be able to formally accept your manuscript and schedule it for publication until you have completed any requested changes.

PRESS

Many congratulations and thank you again for choosing PLOS Biology for publication and supporting Open Access publishing. We look forward to publishing your study. 

Sincerely, 

Ines

--

Ines Alvarez-Garcia, PhD 

Senior Editor 

PLOS Biology
